# DGCyTOF: Deep learning with graphic cluster visualization to predict cell types of single cell mass cytometry data

**Lijun Cheng**[1], **Pratik Karkhanis**[1], **Birkan Gokbag**[1], **Yueze Liu**[2], **Lang Li**[1]*

**1** Department of Biomedical Informatics, College of Medicine, The Ohio State University, Columbus, Ohio, United States of America, **2** The Grainger College of Engineering, The University of Illinois Urbana-Champaign, Urbana and Champaign, Champaign, Illinois, United States of America

* Lang.Li@osumc.edu

**Data Availability Statement:** A Python package (Python 3) and analysis scripts for reproducing the results are available at https://github.com/ lijcheng12/DGCyTOF/. We share all Python source

## Abstract

Single-cell mass cytometry, also known as cytometry by time of flight (CyTOF) is a powerful high-throughput technology that allows analysis of up to 50 protein markers per cell for the quantification and classification of single cells. Traditional manual gating utilized to identify new cell populations has been inadequate, inefficient, unreliable, and difficult to use, and no algorithms to identify both calibration and new cell populations has been well established. A deep learning with graphic cluster (DGCyTOF) visualization is developed as a new integrated embedding visualization approach in identifying canonical and new cell types. The DGCyTOF combines deep-learning classification and hierarchical stable-clustering methods to sequentially build a tri-layer construct for known cell types and the identification of new cell types. First, deep classification learning is constructed to distinguish calibration cell populations from all cells by *softmax* classification assignment under a probability threshold, and graph embedding clustering is then used to identify new cell populations sequentially. In the middle of two-layer, cell labels are automatically adjusted between new and unknown cell populations via a feedback loop using an iteration calibration system to reduce the rate of error in the identification of cell types, and a 3-dimensional (3D) visualization platform is finally developed to display the cell clusters with all cell-population types annotated. Utilizing two benchmark CyTOF databases comprising up to 43 million cells, we compared accuracy and speed in the identification of cell types among DGCyTOF, DeepCyTOF, and other technologies including dimension reduction with clustering, including Principal Component Analysis (*PCA*), Factor Analysis (*FA*), Independent Component Analysis (*ICA*), Isometric Feature Mapping (*Isomap*), t-distributed Stochastic Neighbor Embedding (*t-SNE*), and Uniform Manifold Approximation and Projection (*UMAP*) with *k*-means clustering and Gaussian mixture clustering. We observed the DGCyTOF represents a robust complete learning system with high accuracy, speed and visualization by eight measurement criteria. The DGCyTOF displayed *F-scores* of 0.9921 for CyTOF1 and 0.9992 for CyTOF2 datasets, whereas those scores were only 0.507 and 0.529 for the *t-SNE+k-means*; 0.565 and 0.59, for *UMAP + k-means*. Comparison of DGCyTOF with *t-SNE* and *UMAP* visualization in accuracy demonstrated its approximately 35% superiority in predicting cell types. In addition, observation of cell-population distribution was more intuitive in the 3D visualization in DGCyTOF than *t-*

code for DGCyTOF method, along with a tutorial and demo data, which will guide users through examples (involving CyTOF data). All materials and source codes can be accessed at https://lijcheng12.github.io/DGCyTOF/.

**Funding:** The authors received no specific funding for this work.

**Competing interests:** The authors have declared that no competing interests exists.

*SNE* and *UMAP* visualization. The DGCyTOF model can automatically assign known labels to single cells with high accuracy using deep-learning classification assembling with traditional graph-clustering and dimension-reduction strategies. Guided by a calibration system, the model seeks optimal accuracy balance among calibration cell populations and unknown cell types, yielding a complete and robust learning system that is highly accurate in the identification of cell populations compared to results using other methods in the analysis of single-cell CyTOF data. Application of the DGCyTOF method to identify cell populations could be extended to the analysis of single-cell RNASeq data and other omics data.

## Author summary

Single-cell mass cytometry, also known as cytometry by time of flight is a powerful high-throughput technology that allows analysis of up to 50 protein markers per cell for the quantification and classification of single cells. Traditional manual gating utilized to identify new cell populations has been inadequate, inefficient, unreliable, and difficult to use. A deep learning with graphic cluster (DGCyTOF) visualization is developed in identifying canonical and new cell types. Utilizing two benchmark CyTOF databases comprising up to 43 million cells, we compared accuracy and speed in the identification of cell types among DGCyTOF, DeepCyTOF, and other technologies including dimension reduction with clustering, including Principal Component Analysis, Factor Analysis, Independent Component Analysis, Isometric Feature Mapping, t-distributed Stochastic Neighbor Embedding, and Uniform Manifold Approximation and Projection with $k$-means clustering and Gaussian mixture clustering. We observed that DGCyTOF outperformed all the other methods in accuracy, speed and visualization. The application of the DGCyTOF method to identify cell populations could be extended to the analysis of single-cell RNA-Seq data and other omics data.

This is a *PLOS Computational Biology* Methods paper.

## Introduction

The identification of different cell populations has become an essential focus in cell biology research, and mass cytometry and other such high-throughput technologies are rapidly developing to identify novel cell populations at the level of the individual cell [1–3]. Mass cytometry, also known as cytometry by time of flight, or CyTOF, is a variation of flow cytometry in which protein antibodies are labeled with heavy metal ion tags [4]. Capable of measuring up to 50 proteins in a single cell [5,6] and screening an average of approximately 120 million cells in each experiment, CyTOF serves as an important source of big data and powerful tool for the study of cellular diversity and dynamics. Many recent studies highlight its utility in enabling novel discoveries and enhancing the understanding of cell-to-cell interactions in multiple domains of immunology [7,8] concerning cell-subset heterogeneity and tissue localization [9,10]. Indeed, the characterization of cells by CyTOF improves our understanding of disease progression and drug-response sensitivity or resistance [8].

Many single-cell experiments focus on the identification of the types of cells present in a sample from single-cell CyTOF data, and conventional cell-type identification in CyTOF involves the sequential manual partition, or "gating," of cells into subpopulations. Termed canonical cell population identification, gating requires the visual inspection of scatter plots by one or two protein biomarkers at a time [11–13] using tools such as FlowJo or FlowCore (Yale Flow Cytometry vendor) [14]. Gating is an important step in assigning individual cells into discrete cell types [15], but it is very time-consuming, technically inefficient, and prone to human error. Gating with 50 markers, for example, might yield about $2^{50} = 10^{15}$ cell types [16], so manual gating will not allow an exhaustive search of all cell populations [17]. Furthermore, the high dimensionality and large proportion of none-labeled CyTOF data pose considerable challenges to the identification of cell populations [18,19]. New identification tools are urgently needed that will automate the analysis of data and permit the fulfillment of CyTOF's potential for biological discovery and translational applications.

Many clustering tools can perform this task, which is essential to identify "new" cell populations in explorative experiments. However, relying on clustering is laborious since it often involves manual annotation, which significantly limits the reproducibility of identifying cell-populations across different samples. Many clustering tools can identify cell populations in exploratory experiments [19]. For instance, the *k-means* clustering algorithm iteratively assigns data points (cells) [20] to the nearest centroids (cluster center) and recomputes the centroids based on a predefined number of clusters, and Gaussian Mixture Models (GMMs) [21] tend to group together data points belonging to a single distribution by assuming a certain number of Gaussian distributions, each of which represents a cluster. However, clusters obtained by these algorithms might not be robust. Such algorithms require non-intuitive parameters [22], like the number *k* of clusters. Moreover, for some applications, these algorithms are node-density strategies that might be insufficient to discover the clusters best representing the underlying data structures, such as the node and node relationship of hierarchical structure in a tree or graph [23]. In particular, one large dataset might distribute a large number of very dense objects in some areas and only a few objects in others. Hierarchical density-based spatial clustering of applications with noise clustering (*HDBSCAN*) [24,25] allows the analysis of datasets comprising millions of cells and provides aggregate information on generated hierarchical tree clusters, but information regarding local data structure, that is, single-cell resolution, is sacrificed. Graph-density clustering provides a promising novel strategy for cell-population identification, but the clustering often involves manual annotation, again, a time-intensive and laborious process, and this significantly impedes the reproducibility of cell-population identification across different samples [23]. The learning processes in these clustering methods for the automatic identification of cell populations must depend on prior biological knowledge about the populations to identify canonical cell populations [5]. The *HDBSCAN* clustering method can find arbitrarily-shaped clusters and maintain the topology of the structure of the data and does not require *a priori* specification of the number *k* of clusters in the data, as is the case with *k-means*.

Clustering and dimension-reduction technologies have often been combined to visualize data in two dimensions and thereby enhance interpretive capabilities in the analysis of mass-cytometry data [19,26,27]. Such dimension-reduction techniques include Principal Component Analysis (*PCA*) [28], Factor Analysis (*FA*) [29], Independent Component Analysis (ICA) [30], Isometric Feature Mapping (*Isomap*) [31], t-distributed Stochastic Neighbor Embedding (*t-SNE*) [32], and Uniform Manifold Approximation and Projection (*UMAP*) [33,34]. In particular, the latest *t-SNE* and *UMAP* methods are the two most widely used graph-embedding techniques for visualization. They involve the construction of a high-dimensional graph representation of data followed by optimization of a low-dimensional graph to be as structurally

similar as possible. Both algorithms excel at revealing local structure in high-dimensional data. With lots of clustering and dimension-reduction technologies development, how to select appropriated clustering and dimension reduction in the identification of cell types in CyTOF data remains under investigation.

The recent development of deep learning (*DL*) as a powerful machine-learning method was inspired by the mechanisms of artificial neural networks in the brain, especially those underlying the recognition of patterns from images [35] and the natural processing of language [36]. The deep-learning graphic-clustering approach, DeepCyTOF, shows promise in identification known cell types from massive volumes of CyTOF data by automated labeling technology [37]. DeepCyTOF cell-type clusters are trained using samples labeled with marker genes, assigning target cells to canonical cell types with 99% accuracy compared with the use of real labels in CyTOF data analysis. However, this method cannot detect new cell populations beyond cell types defined by the reference samples and visualization, which is different to DGCyTOF.

DGCyTOF has developed as a new integrated embedding visualization approach in response to the performance of DeepCyTOF in identifying canonical cell types and reducing dimensionality with clustering for visualization in the prediction of new cell types. DGCyTOF combines deep-learning classification and hierarchical stable-clustering methods to sequentially build a tri-layer construct for known cell types, the identification of new cell types, and visualization. At the same time, this technique preserves the local structure among single cells by a detailing of non-linear dimensionality-reduction-based method in 3-dimensional (3D) visualization. DGCyTOF first utilizes marker-labeled samples to calibrate cells into different classifications and then applies dimension-reduction and hierarchical clustering methods to assign the unlabeled samples into the appropriate classifications and calibrate their labels. New cells are identified by the integration of *UMAP* and *HDBSCAN* hierarchical tree representation of the complete data, which preserves the non-linear high-dimensional relationships between cells in low-dimensional space. The integration framework in DGCyTOF allows interactive exploration of the hierarchy by a set of embeddings, 3D scatter plots in which cells are positioned based on the similarity of all marker expressions and used for subsequent analysis, such as the clustering of cells at different levels of the hierarchy. Utilizing two benchmark CyTOF databases comprising up to 43 million cells, we compared accuracy and speed in the precise identification of cell types among DGCyTOF, DeepCyTOF, and other technologies including dimension reduction with clustering, we observed the superior performance of DGCyTOF by eight measurement criteria. In addition, we identified previously missed rare cell populations specifically associated with diseases in both the innate and adaptive immune compartments.

## Materials and methods

### Datasets

Table 1 delineates two CyTOF benchmark data sets, CyTOF1 and CyTOF2 from healthy human bone marrow mononuclear cells (BMMCs) [38,39]. All proteins markers do not have overlapping in the two datasets. (1) The first, CyTOF1, generated by [38] from one healthy volunteer, comprised data of approximately 167,000 cells with 13 markers) about half of which

**Table 1. Two CyTOF benchmark data sets for analysis.**

| Database | No. of Cells | No. of markers | No. of manually gated populations | No. of manually gated cells (label data) |
|---|---|---|---|---|
| CyTOF1 | 167,004 | 13 | 24 | 81,747 |
| CyTOF2 | 265,627 | 32 | 14 | 104,184 |

had been manually gated into 24 cell populations that included protein-expression levels from healthy human BMMCs. (2) The second set, CyTOF2, generated by [39] from two healthy human donors, consisted of almost 266,000 cells with 32 protein-expression markers, about 39% (104,184) of which had been manually gated into 14 major immune cell populations, with the remaining 61% (161,443) labeled as "unassigned."

## Methods

**Overview of DGCyTOF: Deep learning with graphic clustering in calibration-feedback learning for the analysis of CyTOF data.** DGCyTOF combines deep-learning classification, graphic clustering, and dimension reduction in a sequential process to automate the classification of canonical cell populations and thereby overcome many limitations associated with traditional methods of cell-type identification and augment the discovery of novel populations from mass-cytometry data (**Fig 1**). The deep-learning model (**Fig 1B**) is used to predict cell-type labels of a new dataset based on a reference dataset, in which each of the cells had been labeled according to its canonical cell type. The *UMAP* and *HDBSCAN* is used in clustering for new cell type identification by force-directed graph algorithms involving spring-like attractive forces and electrical repulsions between nodes connected by edges on hierarchy clusters that reveal single "outlier" cells in the left small set of CyTOF datasets after canonical cell identification. In the middle layer of classification and clustering, a calibration-feedback system is used to adjust the identification of cell types to reduce false-negative errors (Type II errors) between classification and clustering, and projection of labels in 3D space provides their vivid visualization depiction for easy differentiation of types of cell annotations (**Fig 1A**). The whole process includes the following five steps:

*Step 1*: *Predict canonical cell type using a deep neural network model*: *softmax classifier*. A three-layer artificial neural network constructs the deep classification-learning model for the identification of canonical cell populations (**Fig 1B**). Given a labeled dataset $\{(x_1, y_1), (x_2, y_2), \ldots, (x_m, y_m), \ldots, (x_p, y_p)\}$, where $x_i \in R^n$, $x_i$ are the protein markers, $y_i$ is the labeled cell population, and $y_i \in \{1, \ldots, K\}$, $K$ is the number of manually gated cell populations. In each cell, the *softmax classifier* [40] (multinomial logistic regression) estimates the probabilities of that cell belonging to $K$ different cell populations, and the true population is that with the highest probability (**Fig 1B**). The likelihood of the cell belonging to a particular population is given by $h_\Theta(X)$, which takes the form:

$$h_\Theta(X) = p\big(y = K|x; \theta^{(s)}\big) = x_k^{\theta^{(s)}} = \frac{e^{\theta^{(s)}y_K}}{\sum_{j=1}^{K} e^{\theta^{(s)_j}}} = \frac{e^{s_{y_K}}}{\sum_{j=1}^{K} e^{s_j}} \tag{1}$$

where $\theta^{(s)}$ is the parameter sets on the output layer, $s$, $\theta^{(s)} = \sum_{j=1}^{K} w_j^{(s)} x_j^{(s)} + b^{(s)}$, $w^{(s)} = [w_1^{(s)}, w_2^{(s)}, \ldots, w_j^{(s)}, \ldots w_K^{(s)}]^T$ is the weight parameter, and $b^{(s)}$ is a bias constant. The weight parameter, $w^{(s)}$, and bias, $b^{(s)}$, are learned through training on labeled samples and minimizing the loss function, $L(w, \lambda)$ which is the binary cross-entropy between the observed labels, $Y$, and predictions, $\hat{Y}$ as function (2)

$$L(w, \lambda) = -\frac{1}{K} \sum_{i=1}^{K} [y_i \cdot \log(\hat{y}_i) + (1 - y_i) \cdot \log(1 - \hat{y}_i)] + \lambda \sum_{i=1}^{K} w_i^2 \hat{Y} = f(X, w)$$
$$= w^T X + b \tag{2}$$

where $\lambda$ is the regularization strength (hyperparameter) to penalize "large" weight $w$ coefficients. To output layer, $s$, the $i^{th}$ node ($i$−classification) logistic regression link function in $L$

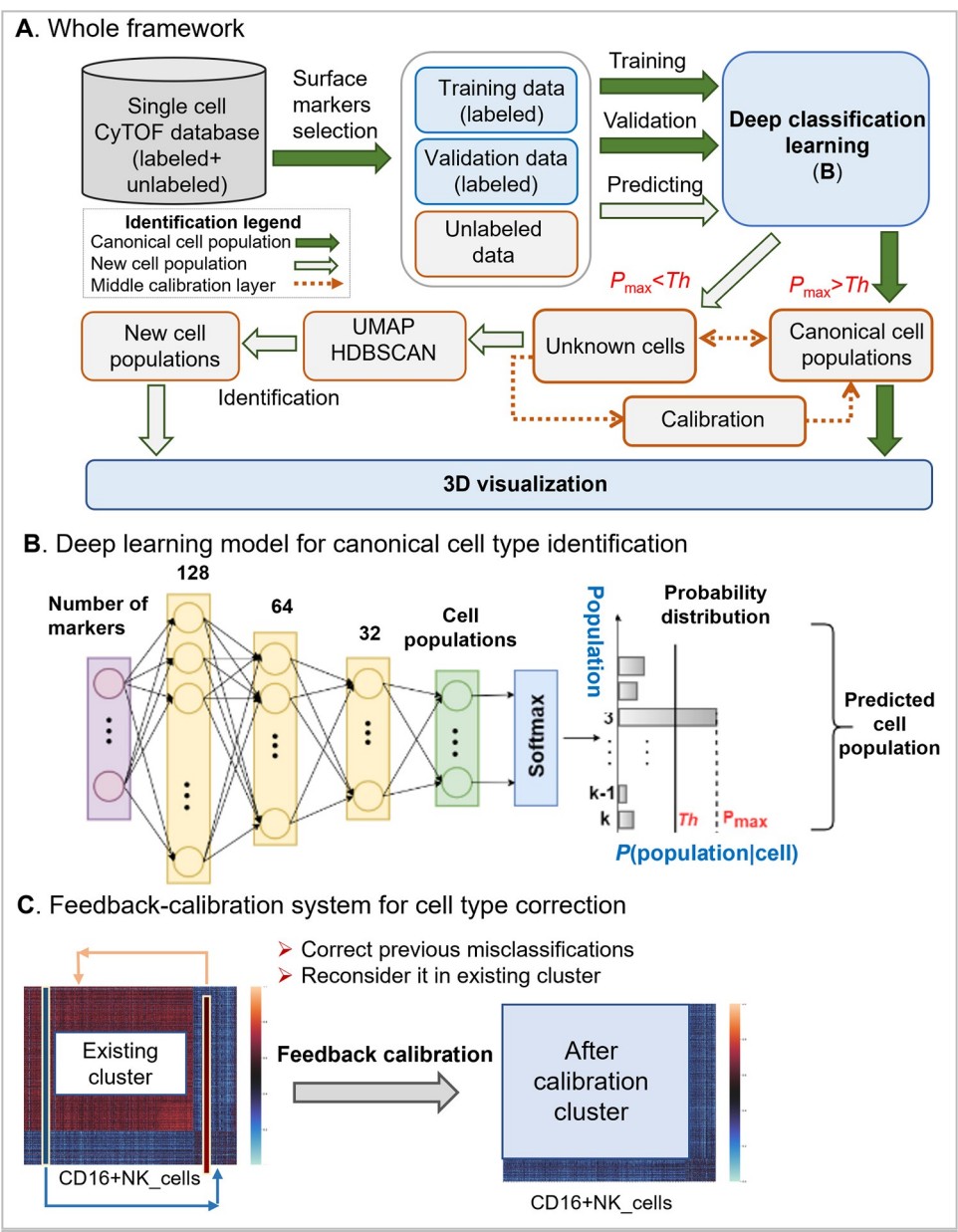

**Fig 1. A framework of DGCyTOF model in the identification of canonical cell population and new cell type populations. A)** The flowchart of DGCyTOF. To single cell data, it includes labeled and unlabeled data in CyTOF database. Identification of cell types includes four processes. (1) To cells labeled, a supervised deep learning automatically identifies canonical cell populations or cell types gated by protein markers, the detailed description sees (**B**). (2) To new cell population, a novel graphic-clustering integrating *UMAP + HDBSCAN* allows a learning of feature representations and preservation of data structure in a network of cell-to-cell interaction for the assignment of clusters for identification of new cell populations. (3) These cell types from classification and clustering are adjusted between (1) and (2) layers above mentioned via a feedback-loop using an iteration calibration system to reduce false-negative errors in the system integrating cell identification. (4) In the final step, a tool permitting three-dimensional (3D) visualization is developed to display the cell clusters, projecting all cell type labels into independent 3D space for their vivid depiction and differentiation to facilitate the identification of cell types. **B)** A three-layer artificial neural network constructs the deep classification-learning model for the identification of canonical cell populations. **C)** A calibration-feedback learning system for cell type correction. After deep learning model in Fig 1A, there are lots of known cell types identified (here called existing cluster). A correlation threshold value averaging the Spearman correlation determines whether the cell belongs to these known cell population. If correlation of the filtered cell with cells from the given canonical cell population is greater than the correlation threshold in this population, we reallocate that cell to this canonical population.

can be written:

$$L_i = -\log\left(\frac{e^{s_{y_i}}}{\sum_{j=1}^{K} e^{s_j}}\right) \tag{3}$$

The loss function, *L*, is the sum over the difference between the observed labels, *Y*, and predictions, $\hat{Y}$, changing Function (2) as follows:

$$L(w, \lambda|\theta^{(s)}) = -\frac{1}{K}\sum_{i=1}^{K}\log\left(\frac{e^{s_{y_i}}}{\sum_{j=1}^{K} e^{s_j}}\right) + \lambda\sum_{i=1}^{K} w_i^2 \tag{4}$$

Adaptive moment (ADAM) optimization algorithm [41] is used to minimize the loss function. The half-half samples labeled is the training data and the test data. After deep learning model is trained, these unlabeled data will input to the trained model to predict their cell-type labels. We thus obtain a prediction probability *matrix* that denotes the probability of each cell belonging to different populations. Rows in the matrix represent populations, *K*, and columns are cells.

To classify cell types, we trained depth-N feed-forward neural nets, each consisting of three **softplus** hidden layers and a **softmax** output layer. To determine the optimal number of layers, penalization weight ($\lambda$) and their drop-out rate in the neural network, DGCyTOF utilized an automated pruning way of building neural network to improve computational and classification performance. By using particle swarm-optimized-based pruning algorithm [42], we selected the optimal multiple parameters (the number of layers, lambda, drop-out rate) in Python during network training. This objective maximization is to keep with the classify accuracy solution that maximizes the area under the *Receiver operating characteristic (ROC)* curve between real labels and prediction labels.

*Step 2*: *Filtering cells to further detect cell populations*. To keep homogeneous cells in a cluster, an optimal threshold value, *Th*, is selected to filter out cells that do not fit well into the cluster. A histogram provides an accurate representation of the probability distribution in each cell population. A cell with a confidence threshold *Th* value below 5% is removed from the labeled cluster from Step1, labeled "unknown," and assigned to a new population for future re-labelling as Eq (5) [5]. By the end of Step 2, about 5% of cells will belong to no particular cell population.

$$\text{assign } x \text{ to } \begin{cases} \arg\ min_{\forall y_i} P(y_i|x); & min\ P(y_i|x) < Th \\ y_{new}\ as\ unknown; & otherwise \end{cases} \tag{5}$$

*Step 3*: *Feedback calibration system to reduce misclassification error*. Despite the strong classification ability of the deep-learning model in Step 1, some cells will be misclassified into incorrect cell-type populations or filtered out of their appropriate clusters, yielding potentially 10 to 20% false negatives after Steps 1 and 2. It is highly desirable to construct calibration feedback and reclassify cells to reduce the error and clarify the homology of cell types within each cluster. Feedback calibration learning system is designed to calibrate the cell-to-cell correlation of inner cell types and improve the homology of cell types within clusters (**Fig 1C**). A Spearman correlation threshold is applied to reallocate these "outlier" points to seek associated clusters [43]. For each canonical cell population, we first calculate the average Spearman correlation of each filtered cell, $\hat{x}_i$, with all cells classified into that particular cell population. Let such correlations with the cells from the cell population $y_j$ be $r_{1j}$, $r_{2j}$,. . .,$r_{qj}$, where *q* is the number of filtered cells, *K* is the number of canonical cell populations, and *j* = 1,. . .,*K*. Let $r_j$ be

the Spearman correlation threshold of cells from the cell population $j$. The threshold $r_j$ is the average Spearman correlation of cells in the cell population $j$. If $r_{qj} > r_j$, then cell $\hat{x}_q \in$ cell-type $j$. In other words, if correlation of the filtered cell with cells from the given canonical cell population is greater than the correlation threshold in this population, we reallocate that cell to this canonical population. If the filtered cell demonstrates correlation greater than the correlation threshold with one or more other canonical cell populations, we reallocate the cell to the population with which it shows highest correlation.

*Step 4*: *Embedding hierarchical tree clustering to detect new cell populations*. Graphic clustering requires a transformation of features from high- to low-dimensional feature space by *UMAP* (uniform manifold approximation and projection for dimension reduction) [33]. The clustering then adopts an explicit approximation of maximum likelihood to estimate the diversity of distribution between the latent representations in the low-dimensional space, at the same time preserving the structure of data in the network of cell-to-cell interactions for the assignment of clusters utilizing HDBSCAN [24].

*Dimension reduction by UMAP. UMAP* [33] is a novel manifold embedding learning technique for general non-linear dimension reduction. Let $\bar{X} = \{x_1, \ldots, x_l\}$ denote cells filtering out of canonical cell populations, where $x_i \in R^n$. *UMAP* works on a weighted graph, denote $G = \{V, E, w\}$ with vertices, edge $E$ with weight $w$. For each cell, *UMAP* first finds its nearest $k$ neighbors by their connection weights, $w$, in the high dimension. The weights $w(i, j)$ depend only on the points in the neighborhoods of $x_i$ and $x_j$ with weights $w_i$ and $w_j$; that is, the weights depend on no more than ($2k+1$) neighbor points. The weight, $w$, is computed from high-dimensional edges for selected neighborhood cells. *UMAP* employs a manifold learning technique to map the connections of each cell and its neighbors to a low-dimensional graph with weight $w'$, which maintains the topology of the global structure of the data and distance between cells in the low-dimensional space [33]. Let $\sigma_i$ be the diameter of the neighborhood of $x_i$, $\rho_i$ be the distance from $x_i$ to its nearest neighbor, and $A$ be the weighted adjacency asymmetric matrix of $G$. We can then make $A$ symmetric by letting:

$$A_{i,j} = w(x_i, x_j) = w_i(x_i, x_j) + w_j(x_j, x_i) - w_i(x_i, x_j)w_j(x_j, x_i) \tag{6}$$

where $w_i\left(x_i, x_j\right) = e^{\frac{-(d(x_i, x_j) - \rho_i)}{\sigma_i}}$, $d$ is the measure of dissimilarity and $\rho_i$ ensures the local connectivity of the manifold.

Given two weights, $w$, $w'$ in the dataset, the cross-entropy $C(w, w')$ between them is:

$$C(w, w') = \sum_{i \sim j} w(i, j) \log\left(\frac{w(i, j)}{w'(i, j)}\right) + (1 - w(i, j))\log\left(\frac{1 - w(i, j)}{1 - w'(i, j)}\right) \tag{7}$$

where $w$ represents the weights computed from high-dimensional filtered cells and $w'$, the weights computed from low-dimensional embedding. *UMAP* optimizes lower-dimensional embedding with respect to cross-entropy by stochastic gradient descent.

*Structure hierarchical tree clustering by HDBSCAN*. A density-based clustering algorithm, *HDBSCAN* constructs a hierarchical tree of clusters and applies a specific stability measure to extract flat clusters from the tree, providing for the discovery of all the small clusters and revealing obvious outliers or noise [24,44–46]. The HDBSCAN algorithm, detailed by McInnes and colleagues [25], can be abstracted into the following steps [45]. 1) It first finds the points in the *epsilon*-neighborhood of every point and applies a mutual reachability distance metric [47] to identify the core points with neighbors of more than "minimum cluster size" (denoted

by *minPts*):

$$d_{mreach-k}(a, b) = max\ core_k(a), core_k(b), d(a, b) \tag{8}$$

where $d(a, b)$ is the distance between cells $a$ and $b$ according to the chosen metric, e.g., Euclidean distance, and $core_k(x)$ is a core distance [45,47]. 2) The algorithm allows representation of the dataset as a graph with data objects as vertices connected by weighted edges with the mutual reachability distances as weights, and the graph is employed to construct a minimum spanning tree (MST). The algorithm then finds the connected components of core points on the neighbor graph, ignoring all non-core points by condensed cluster hierarchy. Sorting the core point edges by mutual reachability distance results in a hierarchical tree structure (dendrogram). 3) By choosing an optimal threshold, *epsilon*, as a global horizontal cut value and selecting all clusters with at least *minPts* points at this density level, we can retrieve the *HDBSCAN* clusters for this *epsilon* from the hierarchy. *HDBSCAN*'s selection algorithm traverses the cluster tree from the bottom up, comparing the stability value of each node to the sum of the stability values of its nested subclusters, thereby propagating and updating stabilities as it ascends the tree until the cluster with the highest stability is found and selected on each tree branch. First, the degree of density of each node is defined by its *epsilon*, or *node stability*, *value*. The resulting clusters more closely approximate the hierarchy of the level sets of the true density distribution of the cells [48]. We employed the HDBSCAN algorithm to discovers clusters of variable densities within node groups with high internal-edge density and the clusters most representative of the underlying structure of data in the low-dimensional space.

During calibration and dimension reduction, Seurat MultiCCA (Canonical Correlation Analysis) method [49] is used to remove the batch effect prior to use DGCyTOF, which was developed in 2017 by the Satija lab. By CCA, the high-throughput data is projected into a subspace to identify correlations across datasets to reduce data dimensionality and capture the most correlated data features to align the data batches.

*Step 5*: *Visualizing cells and cell populations in 3-dimensional space*. For visualization, we first used *UMAP* to project the protein-expression matrix of CyTOF into three dimensions. We added a new label axis, *Z*, to denote cell types previously annotated in Steps 1 and 4. Projecting the similarity, *S*, and associated labels into 3D space enabled the intuitive depiction of hidden structures in the data.

## Algorithm implementation

Implementation of deep learning: The neural network model is implemented in Python using the PyTorch framework [50]. A feedforward neural network comprises three hidden layers, one containing 128 nodes, another, 64 nodes, and the last, 32 nodes (**Fig 1B**), and each layer has a rectified linear unit (*ReLU*) activation function that maps the input vector to non-linear output for the next layer. The output is the *softmax* layer and has the same number of output nodes as the number of cell types. *Softmax* assigns decimal probabilities to each class in our multi-class problem. We used the cross-entropy loss function, optimized using adaptive moment estimation [41], with a learning rate of 0.001 and batch size of 256.

**Other methods and their implementation**: All computation algorithms and their comparison were performed in Python with Scikit-learn [51], a library featuring a wide range of machine-learning algorithms. *UMAP* dimension reduction and HDBSCAN clustering were implemented using their available software [25,33]. The Python system was integrated into the R-Shiny application for visualization. The code and datasets used in this study are available at **https://lijcheng12.github.io/DGCyTOF/**.

## Methodology evaluation measure

**1. Evaluation of dimension-reduction performance.** Three metrics computation time, neighborhood proportion error (*NPE*), and residual variance (*RV*) are used to evaluate performance of dimension-reduction methods–principal component analysis (*PCA*), factor analysis (*FA*), independent component analysis (*ICA*), isometric feature mapping (*Isomap*), t-distributed stochastic neighbor embedding (*t-SNE*), and uniform manifold approximation and projection (*UMAP*).

(1.1) Computation times

All methods were executed on the NVIDIA Tesla P100 GPU cluster provided by the Ohio Supercomputer Center (OSC) to measure time and speed.

(1.2) Neighborhood proportion error

NPE is used to measure total variation distance between the probabilities of cells' assignment to the same subtype in original data and embedding [52]. A smaller NPE indicates better "homology" within a cell type. NPE is defined in Formula (9).

$$NPE = \frac{1}{n}\sum_{i=1}^{n}\delta_{s_i}(P_{s_i}, Q_{s_i}) = \frac{1}{n}\sum_{i=1}^{n}\sum_{k}sup_{a\in[0,1]}|P_s(a) - Q_s(a)| \tag{9}$$

where, where $P_s$ and $Q_s$ represent the empirical density distributions of subtype *s* in the original data and embedding that in low-dimensional space, respectively. The variable *a* represents the fraction of *k* neighbors that belong to the same subtype *s* as the cell in both the original space *P* and low-dimensional embedding *Q*. $\delta_s(P_s, Q_s)$ is the total variation distance [53] for each subtype, $s \in S$, where *S* is the set of all manually gated subtypes.

(1.3) *Residual variance*

Retained variance is a criterion that can be used to choose the appropriate number of principle components to an embedding system in a low dimension space. It represents how much of the information was retained after dimension reduction, such as after *PCA*. It can be evaluated using residual variance (*RV*) by measuring preserved pair-wise distances [31].

$$Residual\ variance = 1 - R^2(D_G, D_y) \tag{10}$$

In Eq (10), $D_G$ is each method's best estimate of intrinsic manifold distances from high dimension space; and $D_y$ is the Euclidean distance matrix in the low-dimensional embedding. *R* is the standard linear correlation coefficient over all entries of $D_G$ and $D_y$. The lower is retained variance, the better of algorithm performance. When no correlation of distances, residual variance attains its maximum value of 1.

**2. Cluster evaluation metrics.** Two metrics are introduced to evaluate the clustering results: adjusted random index (AR**I**) [54] and *V*-measure [55].

**(2.1)** Adjusted Rand index *(ARI)*

The *ARI* metric evaluates whether dimension-reduced similarity cluster results are similar to one other. which is defined by Formula (11) [54]

$$ARI(P^*, P) = \frac{\sum_{ij}\binom{N_{ij}}{2} - \left[\sum_i\binom{N_i}{2}\sum_j\binom{N_j}{2}\right]/\binom{N}{2}}{\frac{1}{2}\left[\sum_i\binom{N_i}{2} + \sum_j\binom{N_j}{2}\right] - \left[\sum_i\binom{N_i}{2}\sum_j\binom{N_j}{2}\right]/\binom{N}{2}} \tag{11}$$

Here, *N* is the number of data points in a given data set and $N_{ij}$ is the number of data points of the class label $C_j^* \in P^*$ assigned to cluster $C_i$ in partition $P^*$. $N_i$ is the number of data points

in cluster $C_i$ of partition $P$, and $N_j$ is the number of data points in class $C_j^*$. In general, an ARI value lies between 0 and 1. The index value is equal to 1 only if a partition is completely identical to the intrinsic structure and close to 0 for a random partition. The advantage of the ARI is that it makes no assumption about the cluster structure. Therefore, it can be used to compare clustering algorithms, such as *k-means* and spectral-clustering algorithms.

**(2.2)** *V-measure*

The *V-measure* is the harmonic mean between homogeneity and completeness as evaluated using a conditional entropy-based external measure of clustering [55]. Homogeneity requires that all clusters contain only cells that belong to a single population, and completeness, that all cells belonging to a given population are elements of the same cluster. Similar to the ARI, the V-measure makes no assumption about the cluster structure.

**3. Evaluation of classification performance.** (3.1) *F-score*

We evaluated our methods using the balanced *F*-score (*F*) as described by Aghaeepour's group [11]. The *F*-score for multiple classes is defined as the weighted average of the $F_i$-score for each cell type:

$$F = \sum_i \frac{C_i}{N} F_i \tag{12}$$

where $C_i$ is the number of cells with type *i*, $N$ is the number of cells, and $F_i$ is the *F*-score for the *ith* cell type versus all other types (including unknown types):

$$F_i = \frac{precision_i \times recall_i}{precision_i + recall_i} \tag{13}$$

*where recall* denotes how many relevant items are selected, and *precision* denotes how many selected items are relevant. The four outcomes can be formulated in a 2×2 contingency **Table 2**:

An *F*-score of 1.0 indicates perfect agreement with the labels obtained by the clustering or classification method.

(3.2) *Receiver operating characteristic (ROC)*

The ROC curve is used as a measure of the quality of classification consistency between the true and prediction labels. The curve is created by plotting the true-positive rate (*TPR*) against the false-positive rate (*FPR*) at various threshold settings, where $TPR = TR/(TP+FN)$, and $FPR = FP/(FP+TN)$, *TP*, *TN*, *FN* and *FP* see contingency **Table 2**. The area of the curve shows the ROC accuracy.

(3.3) *Fowlkes-Mallows score (FMI)*

The FMI is an evaluation metric to evaluate the similarity among clusters obtained after applying different clustering algorithms [55]. The FMI is defined as the geometric mean of pair-wise precision and recall as formular (14).

$$FM = \sqrt{\frac{TP}{TP + FP} \cdot \frac{TP}{TP + FN}} \tag{14}$$

**Table 2. Contingency table for calculating the receiver operating characteristic curve.**

| Total population | Condition positive | Condition negative | Prevalence |
|---|---|---|---|
| Predicted condition positive | TP | FP | *precision = TR/(TP+FP)* |
| Predicted condition negative | FN | TN | |
| | *recall = TR/(TP+FN)* | *specificity = TN/(FP+TN)* | |

Where *TP*, *TN*, *FN* and *FP* see contingency **Table 2**. With a random classification, the FMI will approach zero. A perfect classification will result in an FMI of 1.

## Results

### Identification of cell types by DGCyTOF in two datasets

We used DGCyTOF to automatically identify known and new cell populations in two benchmark CyTOF datasets rresulting in *F-scores* of 0.9921 (CyTOF1) and 0.9992 (CyTOF2) for the identification of labeled cells and cell populations (**Fig 2A and 2C**). **Fig 2B** shows the results of graphic clustering, in which unlabeled cells not belonging to known populations were split into five clusters in the CyTOF1 dataset, and **Fig 2D** shows six clusters of new cell populations among those unknown cells in CyTOF2. **Fig 2 and 2D** depict the final cell populations, including new subtypes and calibration (known) cell types, in 3-dimensional space utilizing the DGCyTOF platform.

### Evaluation and comparison of methods

**External validation of the DGCyTOF model in the identification of known cell types.** External validation is used to evaluate the overfitting phenomenon in the deep-learning algorithm. We randomly separated all labeled samples equally into a training set and a validation set for both CyTOF1 and CyTOF2. We used weight regularization to reduce overfitting of DL model. DGCyTOF model's hyperparameter tuning was done utilizing a regularization $l_1$ normalization to penalize weights sparse to 0 and the top 5% parameters is selected [56]. On the other hand, we designed the dropout and dense strategy for parameters selection to prevent neural networks from overfitting [57,58]. DeepCyTOF selected parameters automatically by its own design. The confusion matrix allows us to visualize the performance of the supervised

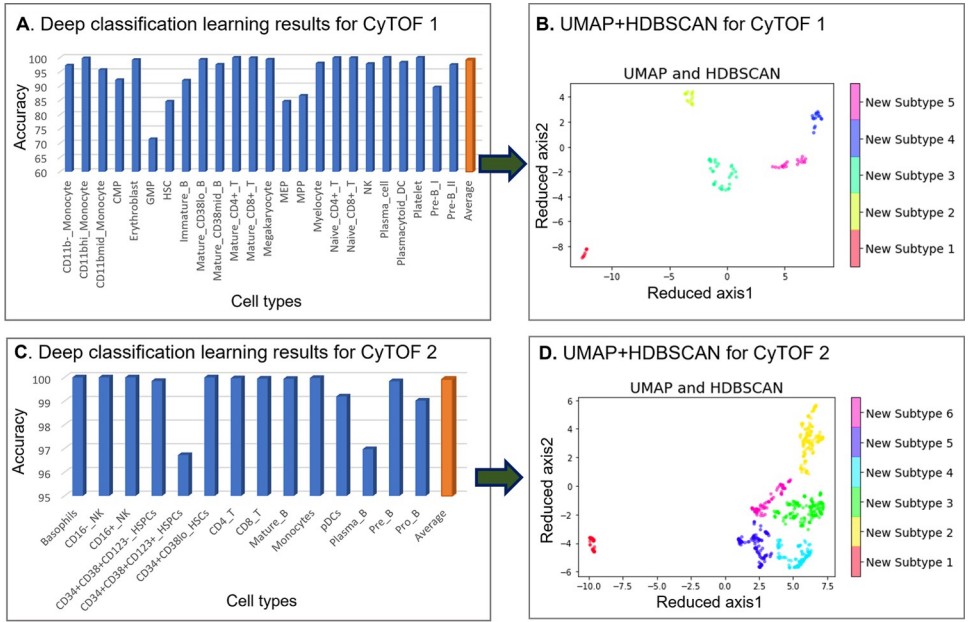

**Fig 2. Cell population identification by DGCyTOF in the analysis of CyTOF1 and CyTOF2 datasets. Fig 2A** identifies the 32 types of known cells by deep classification learning for dataset CyTOF1, and **Fig 2C,** the 13 types of known cells for CyTOF2. **Fig 2B** and **2D** show the spectral clustering for the identification and visualization of unknown cell populations in the two datasets.

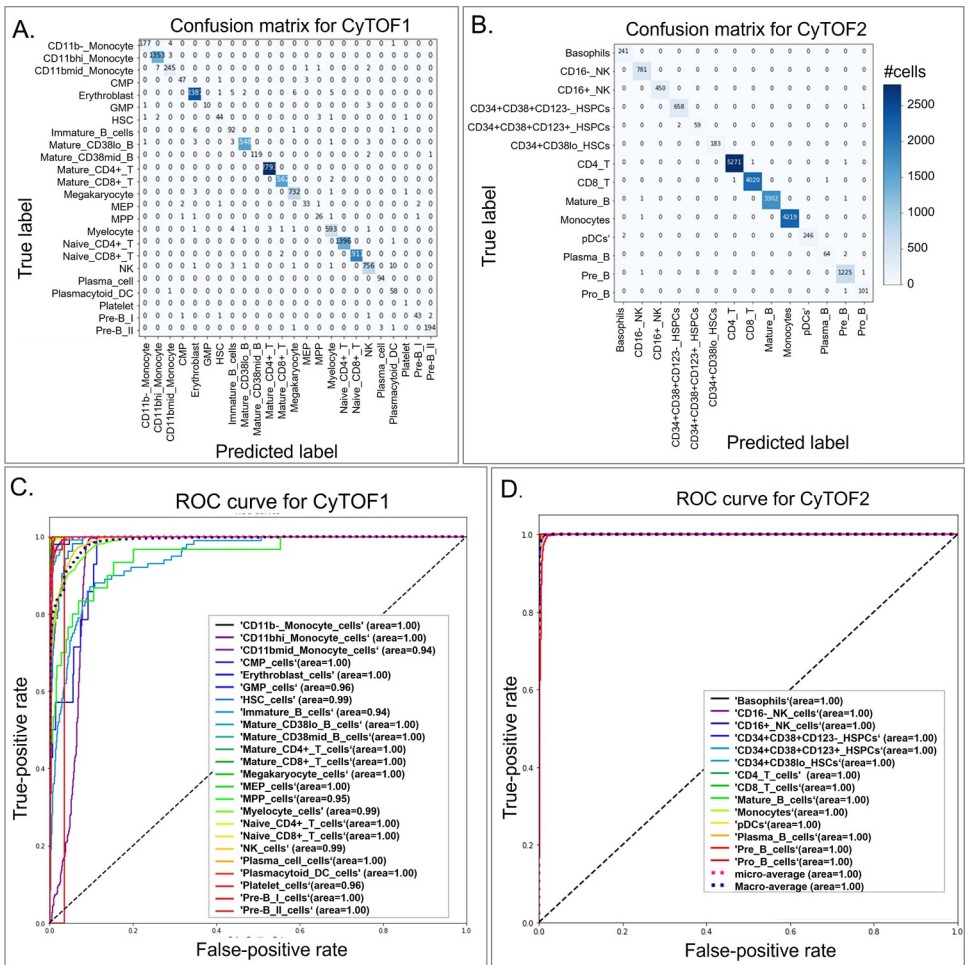

**Fig 3. Comparison of confusion matrices and their associated receiver operating characteristic (ROC) curves for real labels of CyTOF1 and CyTOF2 datasets as assessed utilizing the DGCyTOF model.** (**A**,**B**) confusion matrices for the (**A**) CyTOF1 dataset and (**B**) CyTOF 2 dataset; (**C**,**D**) ROC curves for the (**C**) CyTOF1 dataset and (**D**) CyTOF 2 dataset.

algorithm. Each row of the matrix represents the instances in an actual class, and each column represents the instances in a predicted class (or vice versa). For example, our algorithm correctly assigned 12057 of 12065 cells in our validation data as CD8-T cells and the other eight incorrectly, as CD4-T, CD16-NK, and Plasma-B cells (**Fig 3A and 3B**). We applied a *ROC* metric to evaluate the quality of classifier output using 4-fold cross-validation between the true and prediction labels (**Fig 3C and 3D**). Results showed the average performance in the two testing datasets, with 71.9 to 99.4% accuracy and average *ROC* area of 0.94 to 1.00 in CyTOF1 and 96.5 to 99.8% accuracy and average ROC area of 1.00 in CyTOF2.We compared all prediction results for combination methods with true labels, employing four evaluation criteria to compare the performance of the DGCyTOF and DeepCyTOF methods [37] in classifying labeled data in the CyTOF1 and CyTOF2 databases: *F- score* (harmonic mean of precision and recall) [11]; ARI (adjusted Rand index, a measure of similarity between two clusters) [59]; *FMI* (Fowlkes-Mallows scores) [55]; and *V-measure* (harmonic mean between homogeneity and completeness) [51]. **Table 3** shows the comparison results. All cell-type classifiers used in the DeepCyTOF model were depth-4 feedforward nets, with *softplus* hidden units and a *softmax*

**Table 3. Comparison of methods for averaging performance in the identification of known cell types in training and testing data by different measurements for CyTOF1 and CyTOF2 datasets.**

| Measurement | DGCyTOF | | DeepCyTOF | |
|---|---|---|---|---|
| | CyTOF1 | CyTOF2 | CyTOF1 | CyTOF2 |
| *F–score* ↑ | 0.9921 | 0.9992 | 0.9925 | 0.999 |
| *ARI* ↑ | 0.9924 | 0.9991 | 0.992 | 0.9981 |
| *FMI* ↑ | 0.9932 | 0.9993 | 0.993 | 0.9992 |
| *V-measure* ↑ | 0.9822 | 0.987 | 0.9931 | 0.986 |

Note: All arrow indicators showed good trends

output layer in which hidden layer sizes were set to 12, 6, and 3. The DGCyTOF parameter setting is detailed in the implementation section.

**Evaluation of embedding-clustering performance.** To measure the effectiveness of dimension-reduction techniques that preserve cell proximities, such as cell types, we compared the performances of six techniques with three clustering technologies with respect to computation time, neighborhood proportion error (*NPE*) [52], residual variance (*RV*), and ability to cluster known cell types in two manually gated benchmark mass-cytometry datasets (CyTOF1 and CyTOF2).

We compared these variables among the six techniques–principal component analysis (*PCA*) [28], factor analysis (*FA*) [29], independent component analysis (*ICA*) [30], isometric feature mapping (*Isomap*) [31], t-distributed stochastic neighbor embedding (*t-SNE*) [32] and uniform manifold approximation and projection (*UMAP*) [33]–adding the clustering *K-means*, Gaussian mixture model, and agglomerative clustering algorithm *HDBSCAN* by analysis data in CyTOF1 (13 biomarkers, 24 labeled cell types and CyTOF2 (32 biomarkers, 14 labeled cell types).

We compared all prediction results of the combination methods with respect to the identification of true labels, whereby the *NPE* defined the fraction of cells belonging to a specific subtype under a fixed point, such as the nearest neighbor (*k*), *k* = 20. To evaluate clustering speed and accuracy, we assessed the *F-score* [11], *ARI* [59], *FMI*, and *V-measure* [51,55]. **Tables 4 and 5** show a comprehensive comparison of machine-learning methods by dimension-reduction methods (linear and nonlinear) + clustering methods in our two high-throughput mass-cytometry datasets, in which parts of cell types have been labeled manually. To be comparable, all dimension-reduction methods reduce the dimensionality of a multivariate data to two principal components (2-dimensional embeddings), that can be visualized graphically, with minimal loss of information, here.

In speed comparison, because naive *t-SNE* applications suffer such severe shortcomings as a limited number of cells for analysis and low speed [19,60]. In our experience, the larger the data set, the more severe this problem. Here, we decided to measure the run-time of each of three random sub-sampled datasets CyTOF1 and CyTOF2, each consisting of 10,000 cells, using the average of the three times for each technique as our final computation time comparison. *UMAP* and *t-SNE* are both non-linear graph-based methods and have become an extremely popular technique for visualizing high dimensional data. By these cells, our experiment displays the *UMAP* speed is averaging around 3–4 times faster than *t-SNE*, 18.806 versus 94.466 seconds for CyTOF1 and 16.944 versus 95.609 seconds for CyTOF2 (**Tables 4 and 5**). Other dimension reduction methods *PCA*, *FA*, *ICA* and *Isomap* involved a singular value decomposition (SVD) based on matrix factorization decomposition. Their speed is much fast than other dimension reduction methods with 0.017, 3.618, 0.060, and 0.020 seconds for

**Table 4. Comparison of machine-learning methods by different measurements for CyTOF Dataset 1 (13 biomarkers, 24 labeled cell types).**

| Methods | Measurement | PCA | FA | ICA | Isomap | UMAP—DGCyTOF | t-SNE | NO-reduction |
|---|---|---|---|---|---|---|---|---|
| **Dimension- reduction method** | Computation time (in *seconds*) ↓ | 0.017 | 3.618 | 0.060 | 0.020 | 18.806 | 94.466 | |
| | NPE ↓ | 0.626 | 0.633 | 0.625 | 0.626 | 0.462 | 0.423 | |
| | Retained variance ↓ | 40.6% | 33.4% | - | 40.61% | N/A | N/A | |
| | Visualization ↑ | DD | DD | DD | DD | DD | ED | |
| **k-means clustering** | F-score ↑ | 0.286 | 0.282 | 0.288 | 0.269 | 0.565 | 0.507 | 0.286 |
| | ARI ↑ | 0.236 | 0.228 | 0.236 | 0.216 | 0.556 | 0.483 | 0.235 |
| | FMI ↑ | 0.307 | 0.298 | 0.307 | 0.286 | 0.627 | 0.563 | 0.306 |
| | V-measure ↑ | 0.494 | 0.488 | 0.495 | 0.477 | 0.793 | 0.762 | 0.494 |
| **Gaussian mixture model** | F-score ↑ | 0.305 | 0.304 | 0.505 | 0.285 | 0.588 | 0.502 | 0.530 |
| | ARI ↑ | 0.247 | 0.258 | 0.436 | 0.229 | 0.538 | 0.493 | 0.494 |
| | FMI ↑ | 0.317 | 0.326 | 0.544 | 0.298 | 0.608 | 0.573 | 0.556 |
| | V-measure ↑ | 0.497 | 0.490 | 0.586 | 0.481 | 0.790 | 0.768 | 0.704 |
| *HDBSCAN—DGCyTOF* | F-score ↑ | 0.442 | 0.451 | 0.442 | 0.438 | **0.924** | 0.771 | 0.596 |
| | ARI ↑ | 0.336 | 0.349 | 0.337 | 0.332 | **0.915** | 0.738 | 0.534 |
| | FMI ↑ | 0.526 | 0.531 | 0.524 | 0.524 | **0.925** | 0.789 | 0.621 |
| | V-measure ↑ | 0.557 | 0.570 | 0.552 | 0.555 | **0.905** | 0.850 | 0.557 |

Note–All arrow indicators showed good trends. ARI, adjusted Rand index, measure of similarity between two clusters, involves random labeling independent of the number of clusters; DD, difficult to distinguish; ED, easy to distinguish; FMI, Fowlkes-Mallows score, geometric mean of pair-wise precision and recall; F-score, harmonic mean of precision and recall (values range from 0 [bad] to 1 [good]); NPE, neighborhood proportion error; V-measure, harmonic mean of homogeneity and completeness. All results reflect comparison of two dimensions, and the number of nearest neighbors (k) is 20.

**Table 5. Comparison of machine-learning methods by different measurements for CyTOF Dataset 2 (32 biomarkers, 14 labeled cell types).**

| Methods | Measurement | PCA | FA | ICA | Isomap | UMAP—DGCyTOF | t-SNE | NO-reduction |
|---|---|---|---|---|---|---|---|---|
| **Dimension- reduction method** | Computation time (in *seconds*) ↓ | 0.0253 | 0.208 | 0.054 | 0.021 | 16.944 | 95.609 | |
| | NPE ↓ | 0.536 | 0.525 | 0.535 | 0.536 | 0.399 | 0.393 | |
| | Retained variance ↓ | 31.03% | 27.70% | - | 31.03% | N/A | N/A | |
| | Visualization ↑ | DD | DD | DD | DD | **ED** | ED | |
| **K-means clustering** | F-score ↑ | 0.426 | 0.421 | 0.431 | 0.322 | 0.590 | 0.529 | 0.458 |
| | ARI ↑ | 0.343 | 0.330 | 0.349 | 0.232 | 0.540 | 0.475 | 0.409 |
| | FMI ↑ | 0.444 | 0.431 | 0.448 | 0.340 | 0.637 | 0.578 | 0.537 |
| | V-measure ↑ | 0.609 | 0.582 | 0.611 | 0.444 | 0.799 | 0.744 | 0.735 |
| **Gaussian mixture model** | F-score ↑ | 0.497 | 0.446 | 0.670 | 0.313 | 0.626 | 0.585 | 0.395 |
| | ARI ↑ | 0.406 | 0.353 | 0.573 | 0.221 | 0.577 | 0.534 | 0.339 |
| | FMI ↑ | 0.500 | 0.453 | 0.706 | 0.331 | 0.665 | 0.631 | 0.461 |
| | V-measure ↑ | 0.636 | 0.589 | 0.690 | 0.443 | 0.807 | 0.785 | 0.684 |
| *HDBSCAN—DGCyTOF* | F-score ↑ | 0.669 | 0.650 | 0.665 | 0.560 | **0.981** | 0.923 | 0.684 |
| | ARI ↑ | 0.573 | 0.547 | 0.569 | 0.417 | **0.977** | 0.907 | 0.601 |
| | FMI ↑ | 0.696 | 0.680 | 0.691 | 0.616 | **0.981** | 0.923 | 0.696 |
| | V-measure ↑ | 0.698 | 0.657 | 0.691 | 0.565 | **0.952** | 0.898 | 0.701 |

Note–All arrow indicators showed good trends. ARI, adjusted Rand index, measure of similarity between two clusters, involves random labeling independent of the number of clusters; DD, difficult to distinguish; ED, easy to distinguish; FMI, Fowlkes-Mallows score, geometric mean of pair-wise precision and recall; F-score, harmonic mean of precision and recall (values range from 0 [bad] to 1 [good]); NPE, neighborhood proportion error; V-measure, harmonic mean of homogeneity and completeness. All results reflect comparison of two dimensions, and the number of nearest neighbors (k) is 20.

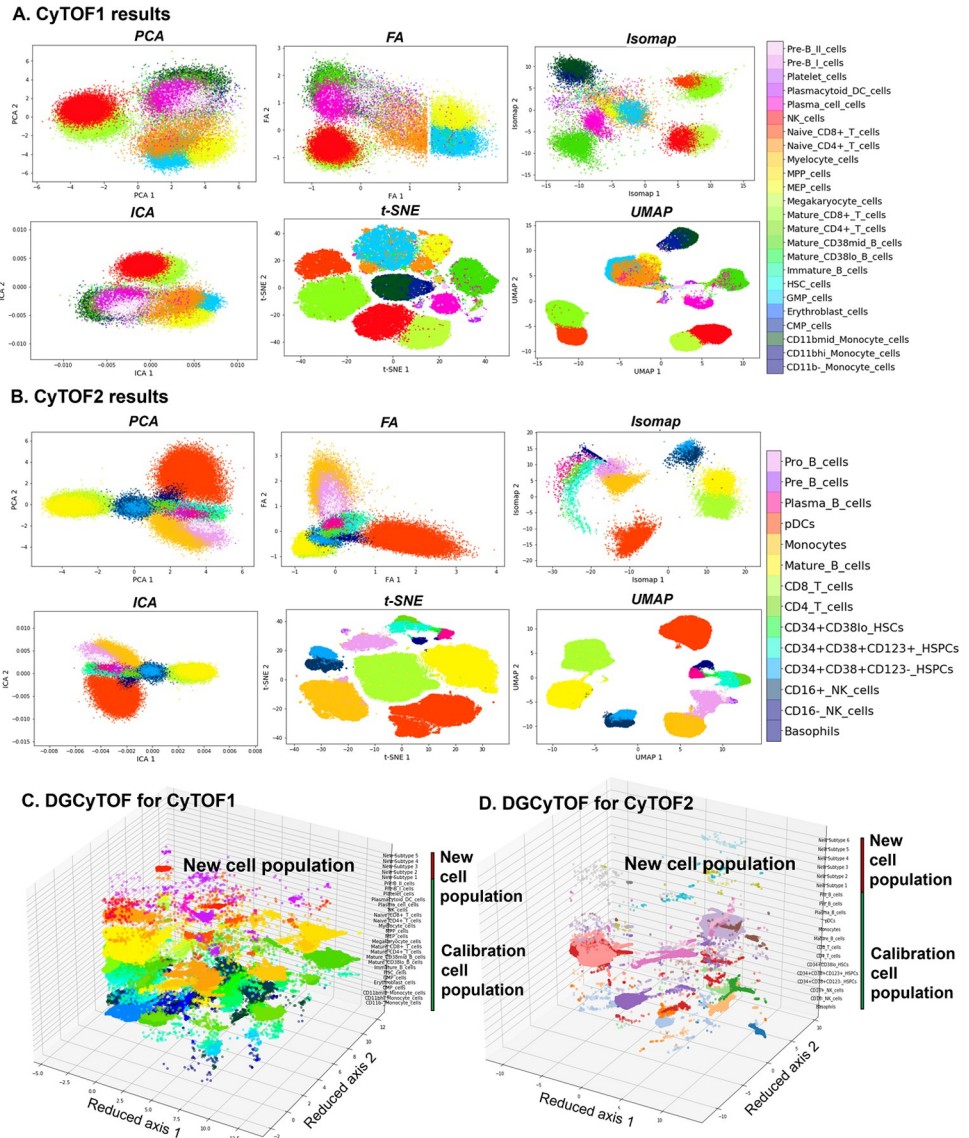

**Fig 4. Visualizations of cell populations in databases CyTOF1 and CyTOF2.** (**A-B**) Two-dimensional visualization of embedding of cells for the identification of dimension-reduction techniques in databases CyTOF1 (**A**) and CyTOF2 (**B**). Cell subtypes are labeled by DGCyTOF for databases CyTOF1 (**C**) and CyTOF2 (**D**).

CyTOF1 and 0.0253, 0.028, 0.054 and 0.021seconds for CyTOF2 (**Tables 4 and 5**). However, they do not fare well in clustering visualization and accuracy in classification.

Accuracy, judged better by a smaller *NPE* and visualization (**Fig 4**), were very similar between the two datasets for *UMAP* and *t-SNE*, with *NPE* variance of 0.462 versus 0.423 for CyTOF1 0.399 versus 0.393 for CyTOF2 0.393. We therefore conclude that graphic models are important technologies that fit well in the analysis of single-cell CyTOF data. The accuracy in identifying cell homology was much higher using the nonlinear structure methods, *UMAP* and *t-SNE*, than any of the other methods, *PCA*, *FA*, *ICA*, and *Isomap*.

In accuracy comparison of dimensional reduction with clustering methods, our graphic clustering HDBSCAN+*UMAP* in DGCyTOF is an extension of the HDBSCAN clustering method in the low-dimension space of *UMAP*. The graphic clustering shows the highest

accuracy in clustering accordance for real labels with F-scores of 0.924 (92.4%) for CyTOF1 data (**Table 4**) and 0.981 (98.1%) for CyTOF2 data (**Table 5**). In contrast, other embedding clustering systems demonstrate average F-scores of 56.45% to 65% (*UMAP+k-means*) and 58.8 to 80% (*UMAP*+Gaussian mixture model). Whereas the F-scores is nearly 0.352 on average to methods *PCA*, *FA*, *ICA*, *Isomap* with or without *k-means* clustering and Gaussian mixture clustering. That means any of the techniques used independently reaches 35.2% accuracy in the identification of cell types except to *t-SNE*. Although *t-SNE+HDBSCAN* improved the performance of clustering accuracy F-scores to 0.771 for CyTOF1 (**Table 4**) and 0.923 for CyTOF2 (**Table 5**). To other metric measurements *ARI*, *FMI* and *V-measure*, we obtained the similar results and conclusions as *F-score*. These results illustrate that DGCyTOF use integration technology HDBSCAN+*UMAP*, which can improve the entire performance of cell identification in CyTOF data by approximately 35%. Overall, DGCyTOF, which integrates *UMAP* and HDBSCAN technology, outperforms two state-of-the-art algorithms, those of the *t-SNE/UMAP*+*k*-means method and *t-SNE*/*UMAP*+Gaussian mixture model. This can be attributed primarily to the requirement of the dimension-reduction method to make fewer assumptions about the nature of cell-to-cell relationships for any given analysis, thereby limiting the accuracy of traditional clustering methods, like the *k-means* method and Gaussian mixture model, for this type of data. In HDBSCAN, an agglomerative clustering algorithm organizes clusters of points into a minimal spanning tree (MST) rather than connecting clusters in a highly connected graph structure in which all connected clusters can create hierarchy relationships between clusters and/or cells in the population. This method sharply improves the accuracy of cell identification. **Fig 4** allows detailed visualization of these phenomena. Another interesting result is that the performance of data dimensionality reduction+clustering is much better than clustering directly without dimension reduction. The result illustrates that dimensionality reduction contributes to main signal extraction during clustering or classification (**Tables 4 and 5**).

**Visualization of cell populations: Clustering and differentiation.**    In 2-dimensional embeddings, researchers are generally interested in the observation of phenotypic clusters and the ease to distinguish them. Phenotypic clustering involves the clustering of cells with the same manually gated subtypes to permit the identification of distinct populations with clear boundaries for known cell types and their differentiation. The visualizations in **Fig 4** show the variation among the different dimension-reduction techniques with respect to their abilities to permit identification of phenotypic clusters and allow easy differentiation. Two-dimensional visualization and cell annotation of the 32-dimensional data of CyTOF 1 (**Fig 4A**) and 12-dimensional data CyTOF 2 (**Fig 4B**). **Fig 4C and 4D** show the results for all 3D visualization utilizing our DGCyTOF tool.

Different dimension-reduction methods combined with different clustering technologies display distinct clustering and observable differentiation patterns. Clusters by *PCA*, *FA*, or *ICA* do not demonstrate the clear separation of cell subtypes; significant mixing between the different populations makes this embedding ineffective for subtype classification. Neither do they clearly distinguish cluster boundaries along which we can observe the process of cell differentiation in CyTOF1 and CyTOF2 data. Clusters by *Isomap* show similarly extensive mixing of different cell subtypes, displaying very little distinct clustering and no observable differentiation patterns. Clusters by *t-SNE*, however, shows the clustering of cell subtypes so that they may be easily distinguished for the differentiation of cells. *UMAP* also demonstrates clearly defined clusters, with large gaps between some distinct groups, and DGCyTOF distinguishes cell subtypes for easy differentiation. Cell classification and clustering in 3D feature space enables clear and easy discernment of each subtype, including newly identified subtypes. From these observations, we conclude that our 3D visualization technique allows the accurate and intuitive

observation of cell-population distribution and detection of cell-to-cell distribution. Visualization using the *t-SNE* method is superior to that of other methods in displaying different cell subtypes, but that in *UMAP* allows ease in distinguishing clusters that enhances differentiation.

**Feedback-calibration simulation between known cell populations and unknown cells.** Feedback-loop learning system is designed to calibrate cell types between inner known-cell type and intra-unknown cell population to improve the inner cell-type homology (**Fig 1C**). Spearman correlation is used to calculate cell-to-cell correlation within clusters, and a threshold value averaging the Spearman correlation determines whether a cell belongs to an unknown cell population. We classified cell-type homology within cells and utilized the feedback-loop learning after deep-learning classification to correct the cell types between known and unknown population. **Fig 1C** illustrates the correction process in CyTOF1 and CyTOF2 data analysis. **Table 6** shows the average Spearman correlation coefficient, *r*, before and after calibration learning and indicates the improvement of many of the classifications of inner cell-type homology (highlighted in bold) after calibration. We can see the cell type homology inner a cluster have an improvement although it is not much.

## Conclusions

Recent advances in mass cytometry (CyTOF) have radically altered the fate of single-cell proteomics by allowing a more accurate understanding of complex biological systems and identify novel cellular subsets [4]. Mass cytometry allows analyses of cells in suspension such as blood but also extended for the analyses of tissue sections. New calculational technology needs in dealing with such a big data to characterize the complex cellular samples' types, where rare cell populations with essential biological function would otherwise be missed. For the first time, we propose DGCyTOF method by integrating the advantages of both the classifier and clusters strategies in identification of known and new cell types according to relative protein abundances from cytometry data. The DGCyTOF method allows automatic and highly accurate assignment of known labels to single cells using deep learning, detects new cell populations utilizing a novel graphic-clustering technique, and employs the guidance of a calibration system to achieve an optimum balance of accuracy. Guided by a feedback calibration system, the model seeks optimal accuracy balance among calibration cell populations and unknown cell types, yielding a complete and robust learning system that is highly accurate in the identification of cell populations compared to results using other methods in the analysis of single-cell CyTOF data.

Compared to other methods in the analysis of two single-cell CyTOF standard datasets, DGCyTOF represents a robust complete learning system with high accuracy and speed in cell-population identification by comparing with popular dimension-reduction techniques *PCA*, *FA*, *ICA*, *Isomap*, *t-SNE* and *UMAP* with *k-means* clustering and Gaussian mixture clustering technology that make minimal assumptions about the nature of relationships between the input cells. We used metric measurements computational *speed*, *ARI*, *FMI*, *F-score*, *NPE*, V-*measure* and *visualization* to assess the quality and utility of reduction in comparison. The DGCyTOF displayed a highly accurate assignment of labels in detecting cell populations. In particular, the DGCyTOF obtains the best performance in the running speed of different algorithm comparison. In addition, observation of cell-population distribution was more intuitive in the 3D visualization in DGCyTOF than *t-SNE* and *UMAP* visualization.

We believe the novel DGCyTOF will place cells into functionally distinct groups and types and allow for detailed analyses of cellular heterogeneity not only for calibration cell types, but for new rare cell types. The DGCyTOF hold great potential to uncover the tissue and immune

**Table 6. Calibration of cell types utilizing calibration feedback for CyTOF1 and CyTOF2 data.**

| CyTOF1 data | Coefficient (*r*) | | CyTOF2 data | Coefficient (*r*) | |
|---|---|---|---|---|---|
| Cell type | Before | After | Cell type | Before | After |
| CD11b-_Monocyte_cells | 0.6627 | 0.6657 | Basophils | 0.6094 | 0.613 |
| CD11bhi_Monocyte_cells | 0.7261 | 0.7261 | CD16-_NK_cells | 0.5474 | 0.5481 |
| CD11bmid_Monocyte_cells | 0.6666 | 0.6696 | CD16+_NK_cells | 0.6138 | 0.617 |
| CMP_cells | 0.4809 | 0.4864 | CD34+CD38+CD123-HSPC | 0.6346 | 0.6403 |
| Erythroblast_cells | 0.3733 | 0.3756 | CD34+CD38+CD123+HSPC | 0.6658 | 0.6992 |
| GMP_cells | 0.5715 | 0.5796 | CD34+CD38lo_HSCs | 0.5879 | 0.5942 |
| HSC_cells | 0.5544 | 0.5734 | CD4_T_cells | 0.6095 | 0.6096 |
| Immature_B_cells | 0.3899 | 0.3932 | CD8_T_cells | 0.6247 | 0.6249 |
| Mature_CD38lo_B_cells | 0.4863 | 0.4866 | Mature_B_cells | 0.6806 | 0.6806 |
| Mature_CD38mid_B_cells | 0.5594 | 0.5614 | Monocytes | 0.6925 | 0.6926 |
| Mature_CD4+_T_cells | 0.5155 | 0.517 | pDCs | 0.6511 | 0.6568 |
| Mature_CD8+_T_cells | 0.5916 | 0.5935 | Plasma_B_cells | 0.6055 | 0.6148 |
| Megakaryocyte_cells | 0.2805 | 0.2854 | Pre_B_cells | 0.6462 | 0.6475 |
| MEP_cells | 0.6374 | 0.6492 | Pro_B_cells | 0.6837 | 0.6914 |
| MPP_cells | 0.4966 | 0.5041 | | | |
| Myelocyte_cells | 0.3919 | 0.3927 | | | |
| Naive_CD4+_T_cells | 0.6915 | 0.6931 | | | |
| Naive_CD8+_T_cells | 0.6891 | 0.6907 | | | |
| NK_cells | 0.4645 | 0.4656 | | | |
| Plasma_cell_cells | 0.4622 | 0.4638 | | | |
| Plasmacytoid_DC_cells | 0.6214 | 0.6388 | | | |
| Platelet_cells | 0.4867 | 0.5078 | | | |
| Pre-B_I_cells | 0.559 | 0.5657 | | | |
| Pre-B_II_cells | 0.5436 | 0.5456 | | | |
| Cell-type homology | 0.537608 | 0.542942 | | 0.632336 | 0.637857 |

system's cellular variation patterns and functionality by these inferring cell types. Application of the DGCyTOF method to identify cell populations could be extended to the analysis of single-cell RNASeq data and other omics data [4,61].

## Author Contributions

**Conceptualization:** Lijun Cheng, Lang Li.

**Data curation:** Lijun Cheng, Yueze Liu.

**Formal analysis:** Pratik Karkhanis, Birkan Gokbag.

**Methodology:** Lijun Cheng, Pratik Karkhanis.

**Resources:** Lang Li.

**Software:** Pratik Karkhanis, Birkan Gokbag.

**Supervision:** Lijun Cheng.

**Validation:** Birkan Gokbag, Yueze Liu.

**Visualization:** Birkan Gokbag.

**Writing – original draft:** Lijun Cheng.

**Writing – review & editing:** Lijun Cheng, Lang Li.

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
