## [Decision Letter · Decision Letter 0]

10 May 2021

Dear Dr. Cheng,

Thank you very much for submitting your manuscript "DGCyTOF:  deep learning with graphic cluster visualization to predict cell types of single cell mass cytometry data" for consideration at PLOS Computational Biology.

As with all papers reviewed by the journal, your manuscript was reviewed by members of the editorial board and by several independent reviewers. In light of the reviews (below this email), we would like to invite the resubmission of a significantly-revised version that takes into account the reviewers' comments.

We cannot make any decision about publication until we have seen the revised manuscript and your response to the reviewers' comments. Your revised manuscript is also likely to be sent to reviewers for further evaluation.

Sincerely,

Kathryn Miller-Jensen, Ph.D.

Associate Editor

PLOS Computational Biology

Jason Haugh

Deputy Editor

PLOS Computational Biology

Reviewer's Responses to Questions

**Comments to the Authors:**

Reviewer #1: In the manuscript titled 'DGCyTOF: deep learning with graphic cluster visualization to predict cell types of single-cell mass cytometry data", the authors created a new pipeline to classify single cells from CyTOF data into known cell types. The pipeline also allows users to identify clusters that cannot fit into known cell types.

While some ideas of the pipeline are novel, such as the calibration step to refine to cell populations, I have many concerns about the proposed method.

1. The authors showed the performance of the classification using two datasets. However, in each experiment, the reported performance is based on training and validation data spit from the same dataset. How will the model perform in the presence of batch effects? The whole purpose of deep-learning is to learn a generalizable model from one labeled dataset and apply the model to classify data from future experiments/studies.

2. An important use of automated clustering is to identify new cell subsets within known cell types. For example, there may be multiple cell subsets within CD8+ T cells, and their proportions could be associated with a disease. Why only limiting the cell clustering to cells with unknown label?

3. While UMAP performs the best when preserving the data's local structure, there will still be information loss during dimension reduction. It is not clear to me why the authors did not choose to perform clustering using original data. Can the author add another column in table 4 and 5 showing clustering performance without dimensionality reduction?

4. The pipeline includes several hyper-parameters, including the number of layers in the DL model, the number of nodes in each layer, the threshold for softmax probability, and the correlation threshold for calibration (not to mention the hyperparameters in HDBSCAN and UMAP). How are the hyperparameters chosen? Is the test dataset used for hyper-parameter selection?

5. The authors shared the package in Github. However, I did not see any documentation other than the installation guide. A concise vignette will be helpful for users to get started.

6. The authors mentioned that "However, relying on clustering is laborious since it often involves manual annotation, which significantly limits the reproducibility of identifying cell-populations across different samples." It is unclear how the proposed clustering method (UMAP + HDBSCAN) improves the reproducibility of identifying cell populations across different samples. Is the clustering stable if applied to individual samples?

Reviewer #2: The authors presented a deep learning framework, termed DGCyTOF, for embedding visualization of single-cell datasets. Specifically, DGCyTOF combines deep learning classification and hierarchical stable-clustering methods to sequentially build a tri-layer construct for known cell types and the identification of new cell types. Based on systematic evaluation, the authors showed a higher performance of the proposed DGCyTOF compared to Principal Component Analysis (PCA), Factor Analysis (FA), Independent Component Analysis (ICA), Isometric Feature Mapping (Isomap), t-distributed Stochastic Neighbor Embedding (t-SNE), and Uniform Manifold Approximation and Projection (UMAP) with k-means clustering and Gaussian mixture clustering. Overall, this is a novel study, which provides a powerful deep learning approach for single-cell data visualization. The manuscript is well-written and the overall computational evaluation is solid. Several specific minor comments should be considered further.

One major concern is hyperparameter tuning for DGCyTOF and other comparing approaches, including PCA, UMAP, t-SNE, and others. The authors are suggested to provide the detailed process for hyperparameter tuning.

As shown figure 3, the authors illustrated AUC =1 for most cell types. The authors are suggested to discuss possible risk of over-fitting.

The authors are suggested to discuss future studies about how to improve biological interpretation of deep learning models.

Finally, computational cost and complexity of DGCyTOF should be compared with other approaches as well, such as PCA, UMAP, t-SNE, and others.

**Have the authors made all data and (if applicable) computational code underlying the findings in their manuscript fully available?**

Reviewer #1: Yes

Reviewer #2: Yes

PLOS authors have the option to publish the peer review history of their article (what does this mean?). If published, this will include your full peer review and any attached files.

Reviewer #1: No

Reviewer #2: No
---

## [Decision Letter · Decision Letter 1]

28 Sep 2021

Dear Dr. Li,

Thank you very much for submitting your manuscript "DGCyTOF:  deep learning with graphic cluster visualization to predict cell types of single cell mass cytometry data" for consideration at PLOS Computational Biology. As with all papers reviewed by the journal, your manuscript was reviewed by members of the editorial board and by several independent reviewers. The reviewers appreciated the attention to an important topic. Based on the reviews, we are likely to accept this manuscript for publication, providing that you modify the manuscript according to the review recommendations.

Sincerely,

Kathryn Miller-Jensen, Ph.D.

Associate Editor

PLOS Computational Biology

Jason Haugh

Deputy Editor

PLOS Computational Biology

[LINK]

Reviewer's Responses to Questions

**Comments to the Authors:**

Reviewer #1: In the revised version of the manuscript titled "DGCyTOF: deep learning with graphic cluster visualization to predict cell types of single-cell mass cytometry data", the authors have addressed most of my concerns by adding new evaluation results and more detailed descriptions of the methods. However, two of my questions remain to be answered.

First, how does DGCyTOF deal with the batch effect? In the point-by-point response, the author has mentioned using CCA (Canonical Correlation Analysis) to adjust for batch effect. However, I can't seem to find the description of CCA in the manuscript.

Second, it is still unclear to me how DGCyTOF is optimized. For example, how does the author determine the number of layers in the neural network and the number of nodes in each layer? The authors also mentioned using l1 regularization and drop-outs. How were the penalization weight (lambda) and the drop-out rates determined? Is there a cross-validation procedure for model optimization?

Reviewer #2: The reviewer appreciated the extensive revision and the authors have addressed all concerns. The reviewer support its publication at PLoS Computational Biology.

**Have the authors made all data and (if applicable) computational code underlying the findings in their manuscript fully available?**

Reviewer #1: Yes

Reviewer #2: Yes

PLOS authors have the option to publish the peer review history of their article (what does this mean?). If published, this will include your full peer review and any attached files.

Reviewer #1: No

Reviewer #2: No

Figure Files:

Data Requirements:

Reproducibility:

References:

---

## [Decision Letter · Decision Letter 2]

16 Dec 2021

Dear Dr. Li,

We are pleased to inform you that your manuscript 'DGCyTOF:  deep learning with graphic cluster visualization to predict cell types of single cell mass cytometry data' has been provisionally accepted for publication in PLOS Computational Biology.

Best regards,

Kathryn Miller-Jensen, Ph.D.

Associate Editor

PLOS Computational Biology

Jason Haugh

Deputy Editor

PLOS Computational Biology

Reviewer's Responses to Questions

**Comments to the Authors:**

Reviewer #1: The authors have answered all my remaining questions.

**Have the authors made all data and (if applicable) computational code underlying the findings in their manuscript fully available?**

Reviewer #1: Yes

PLOS authors have the option to publish the peer review history of their article (what does this mean?). If published, this will include your full peer review and any attached files.

Reviewer #1: No

---

## [Editor Report · Acceptance letter]

29 Mar 2022

PCOMPBIOL-D-21-00432R2 

DGCyTOF:  deep learning with graphic cluster visualization to predict cell types of single cell mass cytometry data

Dear Dr Li,

I am pleased to inform you that your manuscript has been formally accepted for publication in PLOS Computational Biology. Your manuscript is now with our production department and you will be notified of the publication date in due course.

With kind regards,

Anita Estes
